# Attentional Neural Network: Feature Selection Using Cognitive Feedback

**Qian Wang**
Department of Biomedical Engineering
Tsinghua University
Beijing, China 100084
qianwang.thu@gmail.com

**Jiaxing Zhang**
Microsoft Research Asia
5 Danning Road, Haidian District
Beijing, China 100080
jiaxz@microsoft.com

**Sen Song** *
Department of Biomedical Engineering
Tsinghua University
Beijing, China 100084
sen.song@gmail.com

**Zheng Zhang** * [†]
Department of Computer Science
NYU Shanghai
1555 Century Ave, Pudong
Shanghai, China 200122
zz@nyu.edu

## Abstract

Attentional Neural Network is a new framework that integrates top-down cognitive bias and bottom-up feature extraction in one coherent architecture. The top-down influence is especially effective when dealing with high noise or difficult segmentation problems. Our system is modular and extensible. It is also easy to train and cheap to run, and yet can accommodate complex behaviors. We obtain classification accuracy better than or competitive with state of art results on the MNIST variation dataset, and successfully disentangle overlaid digits with high success rates. We view such a general purpose framework as an essential foundation for a larger system emulating the cognitive abilities of the whole brain.

## 1 Introduction

How our visual system achieves robust performance against corruptions is a mystery. Although its performance may degrade, it is capable of performing denoising and segmentation tasks with different levels of difficulties using the same underlying architecture. Consider the first two examples in Figure 1. Digits overlaid over random images are harder to recognize than those over random noise, since pixels in the background images are structured and highly correlated. It is even more challenging if two digits are overlaid altogether, in a benchmark that we call MNIST-2. Yet, with different levels of efforts (and error rates), we are able to recognize these digits for all three cases.

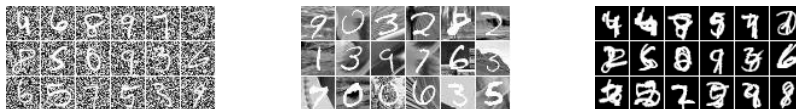

Figure 1: Handwriting digits with different corruptions. From left to right: random background noise, random background images, and MNIST-2

[†]Work partially done while at Microsoft Resarch Asia

Another interesting property of the human visual system is that recognition is fast for low noise level but takes longer for cluttered scenes. Testers perform well on recognition tasks even when the exposure duration is short enough to allow only one feed-forward pass [18], while finding the target in cluttered scenes requires more time[4]. These evidences suggest that our visual system is simultaneously optimized for the common, and over-engineered for the worst. One hypothesis is that, when challenged with high noise, top-down "explanations" propagate downwards via feedback connections, and modulate lower level features in an iterative refinement process[19].

Inspired by these intuitions, we propose a framework called *attentional neural network* (aNN). aNN is composed of a collection of simple modules. The denoising module performs multiplicative feature selection controlled by a top-down cognitive bias, and returns a modified input. The classification module receives inputs from the denoising module and generates assignments. If necessary, multiple proposals can be evaluated and compared to pick the final winner. Although the modules are simple, their combined behaviors can be complex, and new algorithms can be plugged in to rewire the behavior, *e.g.*, a fast pathway for low noise, and an iterative mode for complex problems such as MNIST-2. We have validated the performance of aNN on the MNIST variation dataset. We obtained accuracy better than or competitive to state of art. In the challenging benchmark of MNIST-2, we are able to predict one digit or both digits correctly more than 95% and 44% of the time, respectively. aNN is easy to train and cheap to run. All the modules are trained with known techniques (e.g. sparse RBM and back propagation), and inference takes much fewer rounds of iterations than existing proposals based on generative models.

## 2 Model

aNN deals with two related issues: 1) constructing a segmentation module under the influence of cognitive bias and 2) its application to the challenging task of classifying highly corrupted data. We describe them in turn, and will conclude with a brief description of training methodologies.

### 2.1 Segmentation with cognitive bias

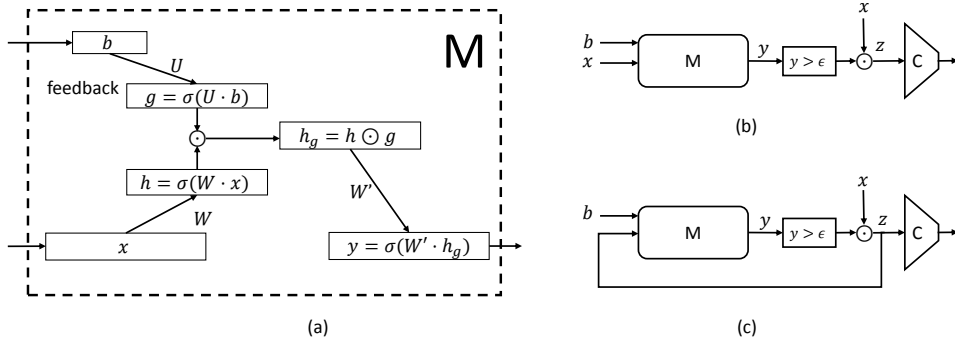

Figure 2: Segmentation module with cognitive bias (a) and classification based on that (b,c).

As illustrated in Figure 2(a), the objective of the segmentation module M is to segment out an object $y$ belonging to one of $N$ classes in the noisy input image $x$. Unlike in the traditional deonising models such as autoencoders, $M$ is given a cognitive bias vector $b \in \{0, 1\}^N$, whose $i$th element indicates a prior belief on the existence of objects belonging to the $i$-th class in the noisy image. During the bottom up pass, input image $x$ is mapped into a feature vector $h = \sigma(W \cdot x)$, where $W$ is the feature weight matrix and $\sigma$ represents element-wise nonlinear Sigmoid function. During the top-down pass, $b$ generates a gating vector $g = \sigma(U \cdot b)$ with the feedback weights $U$. $g$ selects and de-selects the features by modifying hidden activation $h_g = h \odot g$, where $\odot$ means pair-wised multiplication. Reconstruction occurs from $h_g$ by $z = \sigma(W' \cdot h_g)$. In general, bias $b$ can be a probability distribution indicating a mixture of several guesses, but in this paper we only use two simpler scenarios: a binary vector to indicate whether there is a particular object with its associated weights $U_G$, or a *group bias* $b_G$ with equal values for all objects, which indicates the presence of some object in general.

## 2.2 Classification

A simple strategy would be to feed the segmented input $y$ into a classifier $C$. However, this suffers from the loss of details during $M$'s reconstruction and is prone to hallucinations, *i.e.* $y$ transforming to a wrong digit when given a wrong bias. We opted to use the reconstruction $y$ to gate the raw image $x$ with a threshold $\epsilon$ to produce gated image $z = (y > \epsilon) \odot x$ for classification (Figure 2b). To segment complex images, we explored an iterative design that is reminiscent of a recurrent network (Figure 2c). At time step $t$, the input to the segmentation module M is $z_t = (y_{t-1} > \epsilon) \odot x$, and the result $y_t$ is used for the next iteration. Consulting the raw input $x$ each time prevents hallucination. Alternatively, we could feed the intermediate representation $h_g$ to the classifier and such a strategy gives reasonable performance (see section 3.2 group bias subsection), but in general this suffers from loss of modularity.

For iterative classification, we can give the system an initial cognitive bias, and the system produces a series of guesses $b$ and classification results given by $C$. If the guess $b$ is confirmed by the output of $C$, then we consider $b$ as a candidate for the final classification result. A wrong bias $b$ will lead the network to transform $x$ to a different class, but the segmented images with the correct bias is often still better than transformed images under the wrong bias. In the simplest version, we can give initial $b$s over all classes and compare the fitness of the candidates. Such fitness metrics can be the number of iterations it takes $C$ to confirm the guess, the confidence of the confirmation , or a combination of many related factors. For simplicity, we use the entropy of outputs of $C$, but more sophisticated extensions are possible (see section 3.2 making it scalable subsection).

## 2.3 Training the model

We used a shallow network of RBM for the generative model, and autoencoders gave qualitatively similar results. The parameters to be learned include the feature weights $W$ and the feedback weights $U$. The multiplicative nature of feature selection makes learning both $W$ and $U$ simultaneously problematic, and we overcame this problem with a two-step procedure: firstly, $W$ is trained with noisy data in a standalone RBM (i.e. with the feedback disabled); next, we fix $W$ and learn $U$ with the noisy data as input but with clean data as target, using the standard back propagation procedure. This forces $U$ to learn to select relevant features and de-select distractors. We find it helpful to use different noise levels in these two stages. In the results presented below, training $W$ and $U$ uses half and full noise intensity, respectively. In practice, this simple strategy is surprisingly effective (see Section 3). We found it important to use sparsity constraint when learning $W$ to produce local features. Global features (e.g. templates) tend to be activated by noises and data alike, and tend to be de-selected by the feedback weights. We speculate that feature locality might be especially important when compositionality and segmentation is considered. Jointly training the features and the classifier is a tantalizing idea but proves to be difficult in practice as the procedure is iterative and the feedback weights need to be handled. But attempts could be made in this direction in the future to fine-tune performance for a particular task. Another hyper-parameter is the threshold $\epsilon$. We assume that there is a global minimum, and used binary search on a small validation set. [1]

## 3 Results and Analysis

We used the MNIST variation dataset and MNIST-2 to evaluate the effectiveness of our framework. MNIST-2 is composed by overlaying two randomly chosen clean MNIST digits. Unless otherwise stated, we used an off-the-shelf classifier: a 3-layer perceptron with 256 hidden nodes, trained on clean MNIST data with a 1.6% error rate. In the following sections, we will discuss bias-induced feature selection, its application in denosing, segmentation and finally classification.

## 3.1 Effectiveness of feedback

If feature selection is sensitive to the cognitive bias $b$, then a given $b$ should leads to the activation of the corresponding relevant features. In Figure 3(a), we sorted the hidden units by the associated

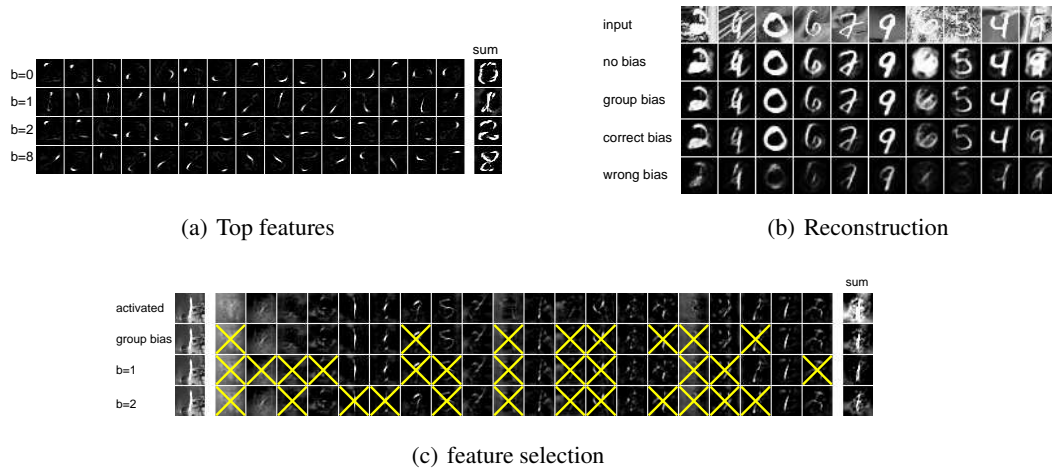

(a) Top features           (b) Reconstruction

(c) feature selection

Figure 3: The effectiveness of bias-controlled feature selection. (a) top features selected by different cognitive bias (0, 1, 2, 8) and their accumulation; (b) denoising without bias, with group bias, correct bias and wrong bias ($b = 1$); (c) how bias selects and de-selects features, the second and the third rows correspond to the correct and wrong bias, respectively.

weights in U for a given bias from the set $\{0, 1, 2, 8\}$, and inspected their associated feature weights in W. The top features, when superimposed, successfully compose a crude version of the target digit.

Since $b$ controls feature selection, it can lead to effective segmentation (shown in Figure 3(b))) By comparing the reconstruction results in the second row without bias, with those in the third and fouth rows (with group bias and correct bias respectively), it is clear that segmentation quality progressively improves. On the other hand, a wrong bias (fifth row) will try to select features to its favor in two ways: selecting features shared with the correct bias, and hallucinating incorrect features by segmenting from the background noises. Figure 3(c) goes further to reveal how feature selection works. The first row shows features for one noisy input, sorted by their activity levels without the bias. Next three rows show their deactivtion by the cognitive biases. The last column shows a reconstructed image using the selected features in this figure. It is clear how a wrong bias fails to produce a reasonable reconstructed image.

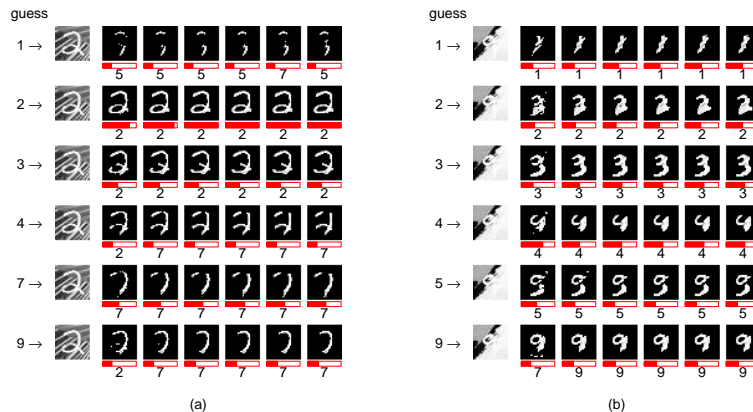

Figure 4: Recurrent segmentation examples in six iterations. In each iteration, the classification result is shown under the reconstructed image, along with the confidence (red bar, the longer the higher confidence).

As described in Section 2, segmentation might take multiple iterations, and each iteration produces a reconstruction that can be processed by an off-the-shelf classifier. Figure 4 shows two cases, with as-

sociated predictions generated by the 3-layer MLP. In the first example (Figure 4(a)), two cognitive biase guesses 2 and 7 are confirmed by the network, and the correct guess 2 has a greater confidence. The second example (Figure 4(b)) illustratess that, under high intensity background, transformations can happen and a hallucinated digit can be "built" from a patch of high intensity region since they can indiscriminately activate features. Such transformations constitute false-positives (i.e. confirming a wrong guess) and pose challenges to classification. More complicated strategies such as local contrast normalization can be used in the future to deal with such cases. This phenomenon is not at all uncommon in everyday life experiences: when truth is flooded with high noises, all interpretations are possible, and each one picks evidence in its favor while ignoring others.

As described in Section 2, we used an entropy confidence metric to select the winner from candidates. The MLP classifier C produces a predicted score for the likelihood of each class, and we take the total confidence as the entropy of the prediction distribution, normalized by its class average obtained under clean data. This confidence metric, as well as the associated classification result, are displayed under each reconstruction. The first example shows that confidence under the right guess (i.e. 2) is higher. On the other hand, the second example shows that, with high noise, confidences of many guesses are equally poor. Furthermore, more iterations often lead to higher confidence, regardless of whether the guess is correct or not. This self-fulfilling process locks predictions to their given biases, instead of differentiating them, which is also a familiar scenario.

## 3.2 Classification

Table 1: Classification performance

| | back-rand | back-image |
|---|---|---|
| RBM | 11.39 | 15.42 |
| imRBM | 10.46 | 16.35 |
| discRBM | 10.29 | 15.56 |
| DBN-3 | 6.73 | 16.31 |
| CAE-2 | 10.90 | 15.50 |
| PGBM | 6.08 | **12.25** |
| sDBN | 4.48 | 14.34 |
| aNN - $\theta_{rand}$ | **3.22** | 22.30 |
| aNN - $\theta_{image}$ | 6.09 | 15.33 |

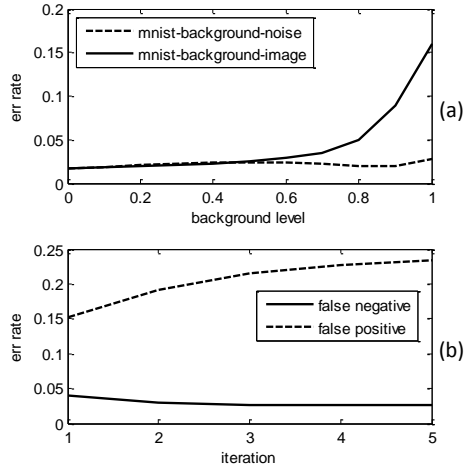

Figure 5: (a) error vs. background level. (b) error vs. iteration number.

To compare with previous results, we used the standard training/testing split (12K/50K) of the MNIST variation set, and results are shown in the Table 1. We ran one-iteration denoising, and then picked the winner by comparing normalized entropies among the candidates, *i.e.* those with biases matching the prediction of the 3-layer MLP classifier. We trained two parameter sets separately in random-noise background ($\theta_{rand}$) and image background dataset($\theta_{image}$). To test transfer abilities, we also applied $\theta_{image}$ to random-noise background data and $\theta_{rand}$ to image background data. On MNIST-back-rand and MNIST-back-image dataset, $\theta_{noise}$ achieves 3.22% and 22.3% err rate respectively, while $\theta_{image}$ achieves 6.09% and 15.33%.

Figure 5(a) shows how the performance deteriorates with increasing noise level. In these experiments, random noise and random images are modulated by scaling down their pixel intensity linearly. Intuitively, at low noise the performance should approach the default accuracy of the classifier C and is indeed the case.

**The effect of iterations**: We have chosen to run only one iteration because under high noise, each guess will insist on picking features to its favor and some hallucination can still occur. With more iterations, false positive rates will rise and false negative rates will decrease, as confidence scores for

both the right and the wrong guesses will keep on improving. This is shown in Figure-5(b). As such, more iterations do not necessarily lead to better performance. In the current model, the predicted class from the previous step is not feed into the next step, and more sophisticated strategies with such an extension might produce better results in the future.

**The power of group bias**: For this benchmark, good performance mostly depends on the quality of segmentation. Therefore, a simpler approach is to denoise with coarse-grained group bias, followed by classification. For $\theta_{image}$, we attached a SVM to the hidden units with $b_G$ turned on, and obtained a 16.2% error rate. However, if we trained a SVM with 60K samples, the error rate drops to 12.1%. This confirms that supervised learning can achieve better performance with more training data.

**Making it scalable**. So far, we enumerate over all the guesses. This is clearly not scalable if number of classes is large. One sensible solution is to first denoise with a group bias $b_G$, and pick top-K candidates from the prediction distribution, and then iterate among them.

Finally, we emphasize that the above results are obtained with only one up-down pass. This is in stark contrast to other generative model based systems. For example, in PGBM [15], each inference takes 25 rounds.

### 3.3 MNIST-2 problem

Compared to corruption by background noises, MNIST-2 is a much more challenging task, even for a human observer. It is a problem of segmentation, not denoising. In fact, such segmentation requires semantic understanding of the object. Knowing which features are task-irrelevant is not sufficient, we need to discover and utilize per-class features. Any denoising architectures only removing task-irrelevant features will fail on such a task without additional mechanisms. In aNN, each bias has its own associated features and explicitly call these features out in the reconstruction phase (modulated by input activations). Meanwhile, its framework permits multiple predictions so it can accommodate such problems.

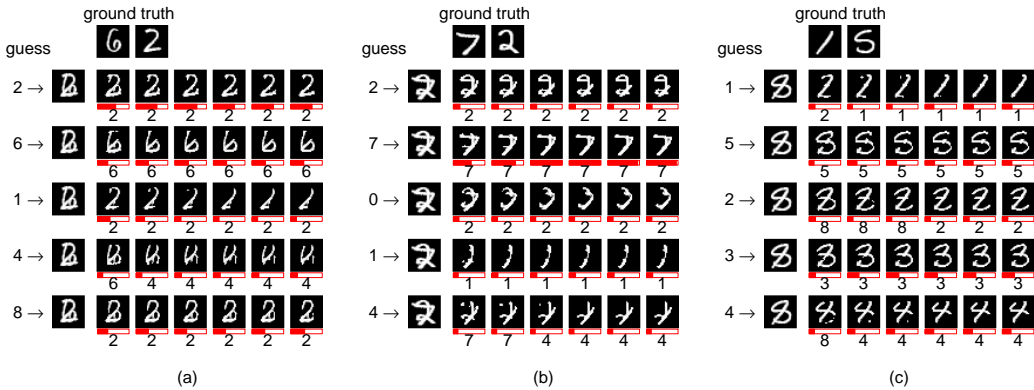

Figure 6: Sample results on MNIST-2. In each example, each column is one iteration. The first two rows are runs with two ground truth digits, others are with wrong biases.

For the MNIST-2 task, we used the same off-the-shelf 3-layer classifier to validate a guess. In the first two examples in Figure 6, the pair of digits in the ground truth is correctly identified. Supplying either digit as the bias successfully segments its features, resulting in imperfect reconstructions that are nonetheless confident enough to win over competing proposals. One would expect that the random nature of MNIST-2 would create much more challenging (and interesting) cases that either defy or confuse any segmentation attempts. This is indeed true. The last example is an overlay of the digit 1 and 5 that look like a perfect 8. Each of the 5 biases successfully segment out their target "digit", and sometimes creatively. It is satisfying to see that a human observer would make similar misjudgements in those cases.

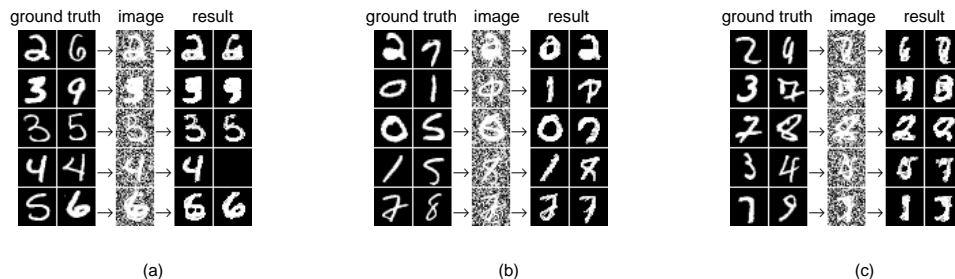

Figure 7: Sample results on MNIST-2 when adding background noises. (a) (b) (c) are examples three groups of results, when both digits, one digit, or none are predicted, respectively.

Out of the 5000 MNIST-2 pairs, there are 95.46% and 44.62% cases where at least one digit or both digits get correctly predicted, respectively. Given the challenging nature of the benchmark, we are surprised by this performance. Contrary to random background dataset, in this problem, more iterations conclusively lead to better performance. The above accuracy is obtained with 5 iterations, and the accuracy for matching both digits will drop to 36.28% if only 1 iteration is used. Even more interestingly, this performance is resilient against background noise (Figure 7), the accuracy only drops slightly (93.72% and 41.66%). The top-down biases allowed us to achieve segmentaion and denoising at the same time.

# 4 Discussion and Related Work

## 4.1 Architecture

Feedforward multilayer neural networks have achieved good performance in many classification tasks in the past few years, notably achieving the best performance in the ImageNet competition in vision([21] [7]). However, they typically give a fixed outcome for each input image, therefore cannot naturally model the influence of cognitive biases and are difficult to incorporate into a larger cognitive framework. The current frontier of vision research is to go beyond object recognition towards image understanding [16]. Inspired by neuroscience research, we believe that an unified module which integrates feedback predictions and interpretations with information from the world is an important step towards this goal.

Generative models have been a popular approach([5, 13]). They are typically based on a probabilistic framework such as Boltzmann Machines and can be stacked into a deep architecture. They have advantages over discriminative models in dealing with object occlusion. In addition, prior knowledge can be easily incorporated in generative models in the forms of latent variables. However, despite the mathematical beauty of a probabilistic framework, this class of models currently suffer from the difficulty of generative learning and have been mostly successful in learning small patches of natural images and objects [17, 22, 13]. In addition, inferring the hidden variables from images is a difficult process and many iterations are typically needed for the model to converge[13, 15]. A recent trend is to first train a DBN or DBM model then turn the model into a discriminative network for classification. This allows for fast recognition but the discriminative network loses the generative ability and cannot combine top-down and bottom-up information.

We sought a simple architecture that can flexibly navigate between discriminative and generative frameworks. This should ideally allow for one-pass quick recognition for images with easy and well-segmented objects, but naturally allow for iteration and influence by cognitive-bias when the need for segmentation arises in corrupted or occluded image settings.

## 4.2 Models of Attention

In the field of computational modeling of attention, many models have been proposed to model the saliency map and used to predict where attention will be deployed and provide fits to eye-tracking data[1]. We are instead more interested in how attentional signals propagating back from higher lev-

els in the visual hierarchy can be merged with bottom up information. Volitional top-down control could update, bias or disambiguate the bottom-up information based on high-level tasks, contextual cues or behavior goals. Computational models incorporating this principle has so far mostly focused on spatial attention [12, 1]. For example, in a pedestrian detection task, it was shown that visual search can be sped up if the search is limited to spatial locations of high prior or posterior probabilities [3]. However, human attention abilities go beyond simple highlighting based on location. For example, the ability to segment and disentangle object based on high level expectations as in the MNIST-2 dataset represents an interesting case. Here, we demonstrate that top-down attention can also be used to segment out relevant parts in a cluttered and entangled scene guided by top-down interpretation, demonstrating that attentional bias can be successfully deployed on a far-more fine-grained level than previous realized.

We have chosen the image-denoising and image-segmentation tasks as our test cases. In the context of image-denoising, feedforward neural networks have been shown to have good performance [6, 20, 11]. However, their work has not included a feedback component and has no generative ability. Several Boltzmann machine based architectures have been proposed[9, 8]. In PGBM, gates on input images are trained to partition such pixel as belonging to objects or backgrounds, which are modeled by two RBMs separately [15]. The gates and the RBM components make up a high-order RBM. However, such a high-order RBM is difficult to train and needs costly iterations during inference. sDBN [17] used a RBM to model the distribution of the hidden layer, and then denoises the hidden layer by Gibbs sampling over the hidden units affected by noise. Besides the complexity of Gibbs sampling, the process of iteratively finding which units are affected by noise is also complicated and costly, as there is a process of Gibbs sampling for each unit. When there are multiple digits appearing in the image as in the case of MNSIT-2, the hidden layer denoising step leads to uncertain results, and the best outcome is an arbitrary choice of one of the mixed digits. a DBM based architecture has also been proposed for modeling attention, but the complexity of learning and inference also makes it difficult to apply in practice [10]. All those works also lack the ability of controlled generation and input reconstruction under the direction of a top-down bias.

In our work, top-down biases influence the processing of feedforward information at two levels. The inputs are gated at the raw image stage by top-down reconstructions. We propose that this might be equivalent to the powerful gating influence of the thalamus in the brain [1, 15]. If the influence of input image is shut off at this stage, then the system can engage in hallucination and might get into a state akin to dreams, as when the thalamic gates are closed. Top-down biases also affect processing at a higher stage of high-level features. We think this might be equivalent to the processing level of V4 in the visual hierarchy. At this level, top-down biases mostly suppresses task-irrelevant features and we have modeled the interactions as multiplicative in accordance with results from neuroscience research [1, 2].

## 4.3 Philosophical Points

The issue of whether top-down connections and iterative processing are useful for object recognition has been a point of hot contention. Early work inspired by Hopfield network and the tradition of probabilistic models based on Gibbs sampling argue for the usefulness of feedback and iteration [14],[13], but results from neuroscience research and recent success by purely feedforward networks argue against it [18],[7]. In our work, we find that feedforward processing is sufficient for good performance on clean digits. Feedback connections play an essential role for digit denoising. However, one pass with a simple cognitive bias towards digits seems to suffice and iteration seems only to confirm the initial bias and does not improve performance. We hypothesize that this "see what you want to see" is a side-effect of our ability to denoise a cluttered scene, as the deep hierarchy possesses the ability to decompose objects into many shareable parts. In the more complex case of MNIST-2, performance does increase with iteration. This suggests that top-down connections and iteration might be particularly important for good performance in the case of cluttered scenes. The architecture we proposed can naturally accommodate all these task requirements simultaneously with essentially no further fine-tuning. We view such a general purpose framework as an essential foundation for a larger system emulating the cognitive abilities of the whole brain.

## Footnotes

*These authors supervised the project jointly and are co-corresponding authors.

[1]The training and testing code can be found in https://github.com/qianwangthu/feedback-nips2014-wq.git

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
