[Reviews · NeurIPS 2014]

Submitted by Assigned_Reviewer_6

Authors propose an attentional neural network that integrates top-down cognitive
bias and bottom-up feature extraction. When there is noise in the stimulus, it can resolve the confusion by setting the appropriate top-down bias. They apply their method for digit recognition on noisy images (random background and random image) as well as two overlaid digits and report high accuracies in par or better than state of the art.

I found this result interesting but I am not quite sure of the novelty of the approach specially considering
a volume of research in recurrent networks, Hopfield nets, predictive coding, etc.

I dont major concerns regarding the approach and results as both seem most likely correct. One concern is though
the generalization of the approach beyond digits to natural scenes where confusion between classes are much more complex.
I think this should be discussed in the paper.

Some minor concerns:

- English needs to be improved. For example, there are several cases where verb does not match with the subject.

- Some more details on dimensionality of vectors in section 2.1 will help the reader.

- Some details on compared models would be helpful.

- Fig. 4: It should be red bar not black bar.

- I dont quite get this: why in Figure 6, last column for input 3, trained NN classifier returns 2 while it clearly looks like a "3"?

Summary: See above.

Submitted by Assigned_Reviewer_13

The authors introduce the Attentional Neural Network which integrates top-down
bias and bottom-up feature extraction in a single architecture.
I like very much the idea of integrating top-down connections, but I have some
concerns on how these have been implemented in the proposed approach and on
the evaluation protocol.

Generally the paper reads well. I strongly recommend to run a spell-check (I
found many typos) and to rewrite few convoluted sentences. Please always favor
linear and short sentences.

The denoising idea to segment out some patterns of interest to be fed to a
classifier is interesting.
It needs to be said that having this gated interaction is not completely novel
and there are already studies on gated RBM and gated Autoencoder.
However, the cognitive bias is a binary N (number of classes) dimensional
vector, and this requires to perform exhaustive search.
Why isn't the bias function of some intermediate representation?

Also, having a separate training for each of the modules makes tricky to assess
the best strategy to "confirm the initial b guess", as also stated in the
paragraph beginning at line 113. This is in my opinion a weak point of the
work which makes it unfeasible for many problems.
I always prefer end-to-end systems, at least here the classifier could have
been trained jointly with the M module, even following the proposed alternating
minimization scheme.

The classification experiment is fine, but it is not clear to me why the
authors did not train the classifier C on the same data. A
baseline is paramount to evaluate the relative improvement of the proposed
approach; how does aNN perform when compared with the simple classifier C
used after M?

The MNIST-2 problem is interesting. The proposed way to disentangle the
various digits is a bit tricky though because of the "validate the guess"
criterion, which is not easy to be selected.
If we are interested in classification, again, what is the performance of C
when taking the first two best guesses (naive) or when C is trained with
cross-entropy to predict the presence of given objects?
Even following the proposed scheme, what if the number of objects is not
known? How to set the criterion? Let's say we were to apply it to a general
object detection benchmark.

Overall I like the approach but there are important experiments which need to
be performed to correctly evaluate the relative improvement of aNN.
Thanks for the work.
Summary: A very interesting approach which definitely will be investigated by many others in the future.
Implementation and evaluation are a bit weak in my opinion and should be completed.

Submitted by Assigned_Reviewer_41

The paper describes a novel method that integrates top-down "cognitive bias" and bottom-up feature extraction. The authors report state-of-the-art results on the MNIST variation dataset.

The main idea of the paper is to propose an integrated framework where bottom-up feature extraction can be affected by a top-down attention process. It is particularly useful in scenarios where the foreground pattern/object of interest is mixed with a cluttered background, or worse, when two different patterns are overlaid.

#quality

Fairly well written paper. Figures of quality, ...

#clarity

1. The authors provide a good introduction and justification of their work.

2. Clear and concise description of the "cognitive bias" module, which is a form of auto-encoder with a path to bias the knowledge of the class present/absent in the input signal.

The reconstructed signal resembles a segmentation mask that highlights the features within the original signal x, that contribute to predicting the class for which the bias is on. That mask is then used to mask irrelevant features from the original signal x, to feed the final classifier.

On: "The hope, however, is that transformation under wrong guess is less perfect than denoised reconstruction under the correct guess.". Wouldn't there be a way of enforcing this during training, with some form of contrastive training? There seems to be a risk with this recursive biasing of the input, to make any prediction possible, irrespective of the original x. I guess a more general question is: how stable is the overall feedback loop?

The experiments section shows appealing examples of successfully reinforced stimuli. Do the authors have examples of less successful results?

#originality/significance

This is definitely original and interesting work. It still feels a bit like early work, with lots of open questions (on stability, dynamics of the system, applicability to more complex signals/modalities, ...), but definitely worth publishing.

Will code be available along with the paper? Very interested in giving it a try!
Summary: The paper describes a novel method that integrates top-down "cognitive bias" and bottom-up feature extraction. The authors report state-of-the-art results on the MNIST variation dataset.
Author Feedback
Author rebuttal: Please make all responses visible to reviewers.
To reviewer 13:
Q: Why isn't the bias function of some intermediate representation?
A: We only used binary vector bias for simplicity in this paper, dimensioned the same as the classification output. For generality, any real-numbered distribution can be used for the bias vector along with more sophisticated search strategies, e.g. tree search. This is an interesting direction of future exploration.

Q: I always prefer end-to-end systems, at least here the classifier could have been trained jointly with the M module, even following the proposed alternating minimization scheme.
A: Thank you for pointing this out. In this paper, we had focused on the module M which carries out attentional processing and opted for a simple and off-the-shelf C for simplicity. Going in the direction of joint training and end-to-end systems, on line 227, we trained a SVM attached to the hidden layer. We will try to jointly train C and M for one-iteration for the revision to see if it improves performance. Joint training for the multiple iteration version is more complicated and will be addressed in future work.

Q: it is not clear to me why the authors did not train the classifier C on the same data. how does aNN perform when compared with the simple classifier C used after M?
A: C works on the denoised image outputted by M, so we trained it on the clean image instead of noisy image. Training a classifier C on the original noisy input gives an error rate of 26.1% and this serves as a baseline performance. When we directly use the C trained on clean data to classify noisy image, the error rate is 85.6%.

Q: The proposed way to disentangle the various digits is a bit tricky though because of the "validate the guess" criterion. What is the performance of C when taking the first two best guesses (naive) or when C is trained with cross-entropy to predict the presence of given objects?
A: We did a few more experiments following your suggestion. When we take from C the first two guesses with highest confidence regardless of whether the guess is validated, the performance is 30.4%/90.9% (both digits/at least one). We were not completely clear of the cross-entropy suggestion.

Q: Even following the proposed scheme, what if the number of objects is not known? How to set the criterion? Let's say we were to apply it to a general object detection benchmark.
A: For object detection, a simple scheme is to use a fixed confidence threshold. We can also imagine more complicated schemes, where we use C on the input image first to come up with top candidates and use them as guesses. The bias vector can also be continuously valued to represent probabilities.

Review 41:
Q: "The hope, however, is that transformation under wrong guess is less perfect than denoised reconstruction under the correct guess." Wouldn't there be a way of enforcing this during training, with some form of contrastive training? how stable is the overall feedback loop?
A: As shown in figure 5, if left to run freely, the feedback loop is not stable and will hallucinate out the biases, so more iterations do not always help the case. This was a surprising result to us. The idea of enforcing the difference during training is an interesting suggestion and we may try it for the revision. We thank the reviewer for this nice suggestion.

Q: Do the authors have examples of less successful results?
A: Some failure cases are provided in figure 4.b and 6.c.

Q: Will code be available along with the paper?
A: We will make it available.

Review 6:
Q: I found this result interesting but I am not quite sure of the novelty of the approach.
A: The main novelty is to combine the various elements into a single extensible framework and demonstrating it on some real-world cases. Details about difference from various previous approaches are given in the discussion section.

Q: the generalization of the approach beyond digits to natural scenes.
A: Dealing with natural scenes will probably require deep architectures, and is probably the most intriguing direction to extend aNN as is. We can start by simply repliacing the encoder and decoder portion of M with a multiple layer convolutional autoencoder and allow attentional interactions only at the top layer of the encoder. Since training for the different components are done separately, such a system should be reasonable to train. More complicated schemes can also be imagined. We will give more suggestions for such possible extensions in the discussion section for the revision.

Q: English needs to be improved. Some more details on dimensionality of vectors in section 2.1 will help the reader. Some details on compared models would be helpful. Fig. 4: It should be red bar not black bar.
A: Suggestions will be followed. We thank the reviewer for the careful reading.

Q: why in Figure 6, last column for input 3, trained NN classifier returns 2 while it clearly looks like a "3"?
A: This is a case of classifier failure. By running more iterations, we found that the middle stick feature in 3 is important for its correct classification, which is not obviously present in that particular example. However, using an off-the-shelf CNN as C dealt with this case correctly at the 2nd iteration. We will also present this result in the new revision.